# SESN3 Inhibited SMAD3 to Relieve Its Suppression for *MiR-124*, Thus Regulating Pre-Adipocyte Adipogenesis

**DOI:** 10.3390/genes12121852

**Published:** 2021-11-23

**Authors:** Weimin Lin, Jindi Zhao, Mengting Yan, Xuexin Li, Kai Yang, Wei Wei, Lifan Zhang, Jie Chen

**Affiliations:** College of Animal Science and Technology, Nanjing Agricultural University, Nanjing 210095, China; 2018205002@njau.edu.cn (W.L.); 2019105005@stu.njau.edu.cn (J.Z.); 2018105035@njau.edu.cn (M.Y.); 2020105004@stu.njau.edu.cn (X.L.); 2019105037@stu.njau.edu.cn (K.Y.); wei-wei-4213@njau.edu.cn (W.W.); lifanzhang@njau.edu.cn (L.Z.)

**Keywords:** SESN3, SMAD3, *ssc-miR-124-3p*, pig, adipogenesis

## Abstract

Sestrin-3, together with the other two members Sestrin-1 and Sestrin-2, belongs to the Sestrin family. The Sestrin protein family has been demonstrated to be involved in antioxidative, metabolic homeostasis, and even the development of nonalcoholic steatohepatitis (NASH). However, the adipogenic regulatory role of SESN3 in adipogenesis still needs to be further explored. In this study, we demonstrated SESN3 inhibited porcine pre-adipocyte proliferation, thus suppressing its adipogenesis. Meanwhile, SESN3 has been demonstrated to inhibit Smad3 thus protecting against NASH. Further, for our previous study, we found *mmu-miR-124* involved in 3T3-L1 cell adipogenesis regulation. In this study, we also identified that *ssc-miR-124* inhibited porcine pre-adipocyte proliferation, thus suppressing its adipogenesis, and the SMAD3 was an inhibitor of *ssc-miR-124* by binding to its promoter. Furthermore, the *ssc-miR-124* targeted porcine *C/EBPα* and *GR* and thus inhibited pre-adipocyte adipogenesis. In conclusion, SESN3 inhibited SMAD3, thus improving *ssc-miR124*, and then suppressed *C/EBPα* and *GR* to regulate pre-adipocytes adipogenesis.

## 1. Introduction

Overweight and obesity lead to a batch of diseases including but not limited to metabolic disease, hypertension, cardiovascular disease, and type II diabetes (T2D), which becomes one of the major threats to human health [1,2]. As a basic unit of adipose tissue, adipocyte differentiation and proliferation lead to adipose tissue expansion, and excessive adipose tissue cause obesity-related metabolic syndromes. Hence, adipocytes are emerging as a significant target in the treatment of obesity-related metabolic syndromes in the clinical setting [3]. Meanwhile, the molecular mechanisms of adipocyte differentiation and proliferation are of significant and escalating biomedical interest.

According to the classical extensive consensus on adipogenesis, hundreds of factors and genes involve enormous and complicated interaction networks to control this progress. Among them is the peroxisome proliferator activated receptor (PPAR) family, containing PPARα and PPARβ/δ, which are numbers of the nuclear receptor superfamily of ligand-activated transcription factors, and important adipogenic regulators [4]. In particular, PPARγ has been widely confirmed to play a critical role in adipocyte development, such as insulin signaling pathway, which is involved in regulating insulin sensitivity [5,6,7]. Apart from that, the CCAAT/enhancer binding protein (C/EBP) family is another key adipogenic regulator, including C/EBPα/β/γ/δ/ε and ζ, which can bind to the promoter of target genes through the CCAAT element, and form both homodimers and heterodimers to regulate their targets’ expression level [8,9]. Moreover, C/EBPα and PPARγ activate with each other in mammalian cells to co-regulate adipogenesis. Meanwhile, it has been proven that there is a broad overlap in both of their transcriptional target genes [10]. The C/EBPα and PPARγ mainly played regulatory role within the early and middle stage of adipogenesis. However, for the middle to late stage of adipogenesis, the fusing of small lipid droplets to mature lipid droplets causes the expansion of adipocytes. Numerous regulators are involved in this progress, including glucose transporter 4 (GLUT4), lipoprotein lipase (LPL), stearyl-CoA-desaturase (SCD) and fatty acid synthase (FAS), and so on [11,12].

MicroRNA (miRNA) is an important part of ncRNAs that has been demonstrated involving a series of biological progress, including adipogenesis. Generally, miRNAs are a kind of small molecular with ~21 nucleotides ncRNA, by targeting in either 3′untranslated region (3′UTR) of coding genes, or ncRNA, for instance, long non-coding RNAs (lncRNAs), circleRNAs (circRNA) and pseudogenes, to play the post-transcriptional role [13,14,15]. A series of miRNAs have been proven involving adipogenesis regulation. For instance, *miR-27* inhibited lipoprotein lipase (LPL) [16], and *miR-130* suppressed adipocytes adipogenesis by targeting *PPARγ* [17]. *MiR-103* was one of miRNAs that improved 3T3-L cell differentiation by targeting *MEF2D* and activating the Akt/mTOR signal pathway [18]. The other adipogenic miRNAs contained but was not limited to *miR-34* [19], *miR-17* [20], *miR-124* [21], *miR-144* [22], and so on.

Sestrin (SESN) proteins have been demonstrated as multifunctional ones by their biochemical characterization. Among them, Sestrin 1 and 2 share the conserved domain structure, whereas the structure of Sestrin 3 differs from other eukaryotic proteins, including Sestrin 1 and 2 [23]. Sestrin is able to activate adenosine monophosphate-activated protein kinase (AMPK) and the mammalian target of rapamycin kinase complex 2 (mTORC2,) but inhibits mammalian target of rapamycin kinase complex 1 (mTORC1) [24]. SESN1 and SESN2 are regarded as the sensor to leucine for the mTORC1 regulation [25,26]. Though SESN1/2/3 shared part common functions in AMPK and mTOR regulation, whereas there is a hierarchy of their leucine binding ability: Generally, SESN1 is the highest, while SESN3 is the lowest [24]. Moreover, SESN3 has been demonstrated to protect against diet-induced non-alcoholic steatohepatitis (NASH) in mice by directly inhibiting SMAD3 through protein–protein interaction, thus suppressing the TGFβ-SMAD3 signal pathway [24]. However, the role of SESN3 in adipogenesis still needs to be further explored. Hence, in this paper, we tried to study the adipogenic role of SESN3 in pre-adipocytes adipogenesis.

## 2. Materials and Methods

### 2.1. Experiment Animals

The animals in this study were 7-days-old Erhualian piglets, and they all came from the Changzhou Erhualian Pig Production Cooperation (Changzhou, Jiangsu, China).

All animal experiments including the pre-adipocyte collection were approved and reviewed by the Animal Ethics Committee of Nanjing Agricultural University (STYK (Su) 2011-0036).

### 2.2. Cell Culture, Transfection and Differentiation

Adipose tissue, isolated from the porcine back subcutaneous, was cut with scissors into 1-mm^3^ pieces, and we used 1 mg/mL of collagenase type I (Invitrogen, Carlsbad, CA, United States). The sample was kept in a 37 °C, 50 rpm/min shaking bath for over 2 h to digest. We added growth medium (89% Dulbecco’s modified Eagle’s medium/Ham’s F-12 (DMEM-F12), 10% fetal bovine serum and 1% penicillin-streptomycin) 1.5 times the volume of the total digestion product to stop the digestion progress. The pre-adipocytes were collected with a 200-mm nylon mesh, filtrated, and were obtained via differential centrifugation. The pre-adipocytes were cultured in the growth medium at 37 °C with 5% CO_2_. The medium was replaced with a fresh one every 2 days.

When a density of 85% confluence, the cell was cultured in 6- or 12-well plates, following the protocol transfection purpose plasmids or oligonucleotides into the cell using Lipofectamine 3000 (Invitrogen, Shanghai, China). The plasmids and oligonucleotides are shown in Appendix A.

The DIM inducer comprised of 2.5 mM of dexamethasone, 8.6 mM of insulin, 0.1 mM of 3-isobutyl-1 methylxanthine (IBMX), 1% penicillin-streptomycin, and 10% FBS in Dulbecco’s modified Eagle’s medium/Ham’s-high glucose (DMEM-HG) (Sigma–Aldrich, Shanghai, China) and was used to induce for adipogenic differentiation. After 4 days’ induction of adipogenic differentiation, the medium was replaced with maintenance medium comprised of 8.9 mM of insulin and 10% FBS-DMEM-HG until day 8. Every 2 days, we replaced it with a fresh medium.

### 2.3. RNA Isolation, Library Preparation, and RT-PCR

We used the Trizol reagent (TaKaRa, Dalian, China) to isolate the total RNA of the cell or tissue. The cDNA libraries of mRNA and miRNA were either reverse-transcribed by the PrimeScript^TM^ RT Master Mix (TaKaRa, Dalian, China), or miRNA 1st Strand cDNA Synthesis Kit (by stem-loop) (Vazyme, Nanjing, China), respectively. The AceQ Universal SYBR qPCR Master Mix (Vazyme, Nanjing, China) and the miRNA Universal SYBR qPCR Master Mix (Vazyme, Nanjing, China) were used to detect Quantitative real-time PCR (q-PCR) of mRNA and miRNA, respectively. The relative level of RNA expression was normalized to *GAPDH* and miR-17 [27,28,29] expression levels using the 2^−ΔΔCt^ method. Every sample was detected in triplicate. Primers were shown in Appendix A.

### 2.4. Oil Red O Staining and Triglyceride Assay

We used 4% paraformaldehyde (Biosharp, Hefei, China) to fix differentiated porcine pre-adipocytes for 30 min after gently washing three times with fresh 1× PBS. We washed the fixed cells three times with 1× PBS, using the 60% saturated oil red O to stain for 30 min (Sigma–Aldrich, Shanghai, China).

Subsequently, we captured images of the cells using a Zeiss Axiovert 40 CFL inverted microscope (Thornwood, NY, United States).

Quantitation of total triglyceride was carried out via eluting oil red O in pure isopropanol and absorbance was measured at the 510-nm wavelength.

### 2.5. Cell Counting Kit-8 (CCK-8) Assay

Pre-adipocytes were seeded in a 96-well plate for 24 h, adding 10 μL of CCK-8 reagent (UE, Shanghai, China) to each well and incubating the for 4 h in the cell culture incubator. The absorbance was measured at 450 nm with a microplate reader. The cell viability was calculated with the following equation: (OD treatment—OD blank). Each group had nine independent replicates (*n* = 9).

### 2.6. Cell Division Assay by EdU Incubation

The pre-adipocytes were incubated for 2 h with YF^®^ 488 Click-iT EdU (50 μM) (UE, Shanghai, China). Then, we washed the pre-adipocytes with 1× PBS, fixed them in 4% paraformaldehyde for 30 min, incubated them with 2 mg/mL glycine solution to neutralize the residual paraformaldehyde for 5 min, washed them with 3% BSA (configurating with 1× PBS) thrice, and then incubated them with 0.5% Triton X-100 for 20 min. Finally, we washed them with 3% BSA (configurating with 1× PBS) thrice and incubated them with Click-iT EdU work solution following protocol for 30 min. Subsequently, we washed them with 1× PBS thrice, finally incubating with 1× Hoechst33342 for 30 min. The representative images were captured via confocal microscopy (LSM700META, Zeiss, Oberkochen, Germany).

### 2.7. Luciferase Reporter Assay

The porcine pre-adipocytes were cultured in 12- or 24-well plates until the density was of 85% confluence. The pGL3 basic vector either contained the promoter of *Sesn3* or *ssc-miR-124* transfected into pre-adipocytes, by using Lipofectamine 3000 (Invitrogen Shanghai, China). Furthermore, the pmirGLO vector contained the 3′UTR of *GR* gene, which co-transfected with *ssc-miR-124* mimics/NC or *ssc-miR-124* inhibitors/NC, respectively.

The Dual-Luciferase Reporter Assay System (Promega, Madison, WI, United States) was used to quantify Firefly and Renilla luciferase activity.

### 2.8. Chromatin Immunoprecipitation Assay PCR (ChIP-PCR)

We used the ChIP Assay Kit (Boytime, Nanjing, China) following the manufacturer’s instruction to detect FoxO1 protein binding with the promoter of *Sesn* and SMAD3 binding to the promoter of *ssc-miR-124*, respectively. Briefly, 1% formaldehyde was used to cross-link pre-adipocytes at 37 °C for 10 min, quenching the cross-linking by 1× glycine for 5 min at room temperature. We used the ultrasonic cell smash machine VCX750 (Sonics, United States) to smash the pre-adipocytes and to obtain DNA fragments between 200 and 1000 bp as verified by ethidium bromide electrophoresis. For immunoprecipitation, 5 μg of antibody against FoxO1 (cat 383312 ZEN BIO) and SMAD3 (bs-3484R, Bioss) was incubated with 100 mL of cellular lysis at 4 °C overnight. Protein A Agarose/SalmonSperm DNA at was incubated at 4 °C for 2 h to isolate the immunoprecipitated complexes, and then washed as following: low salt wash buffer, high salt wash buffer, LiCl wash buffer once, and TE buffer twice. The final PCR analysis subjected to ChIP DNA and specific primers are shown in Appendix A.

### 2.9. Western Blotting

Briefly, total protein was extracted from porcine pre-adipocytes by using radioimmunoprecipitation assay (RIPA) lysis buffer (Beyotime, Jiangsu, China), following the protocol, and then quantified the protein by BCA Protein Assay kit (Beyotime, Jiangsu, China). We loaded the protein sample (1.5 μg/well) into a 12% SDS-PAGE gel (Zoman, Beijing, China). After its electrophoresis, SDS-PAGE was transferred to a PVDF membrane (Millipore, Billerica, MA, United States). Subsequently, we used the 1× TBST containing 5% bovine serum albumin (BSA) to block the transferred membranes for 2 h at room temperature, followed by overnight primary antibody incubation (ZenBioScience, Chengdu, China) at 4 °C. On the second day, the immunoblot membranes were washed thrice with 1× TBST for 30 min, and horseradish peroxidase conjugated secondary antibody was used to incubate for 2 h at room temperature. Using the ECL Chemiluminescence Detection Kit (Vazyme, Nanjing, China) to develop the blots, and using the VersaDoc 4000 MP system (Bio-Rad) to photograph.

### 2.10. Bioinformatics Analysis

Four kinds of online software were used to predict the *ssc-miR-124* targets, respectively, including miRanda (http://cbio.mskcc.org/microrna_data/miRanda-aug2010.tar.gz, accessed on 27 March 2010), miRmap (https://mirmap.ezlab.org/, accessed on 9 January 2013), TargetScan (http://www.targetscan.org/, accessed on 15 March 2018), and PITA (http://genie.weizmann.ac.il/pubs/mir07/mir07_data.html, accessed on 23 September 2007). RNAhybird (http://bibiserv.techfak.uni-bielefeld.de/rnahybrid/submission.html, accessed on 18 September 2017) was used to predict the binding site and affinity between C/EBPα, GR, and ssc-miR-124. The precursor and mature sequences of ssc-miR-124 were obtained from miRBase (http://www.mirbase.org/, accessed on 7 June 2018).

PromoterScan 2 (http://bimas.dcrt.nih.gov:80/molbio/proscan, accessed on 4 February 1999) and JASPAR (http://jaspar.genereg.net/, accessed on 1 January 2013) were used for promoter and transcription factor prediction. Image J was used to quantify the results of C/EBPα and GR Western blotting, and the relative fluorescence intensity (EdU/DAPI).

### 2.11. Statistical Analysis

SPSS software (21.0 version, IBM, United States) was used to carry out the statistical analysis. All data were presented as means ± standard error of mean (s.e.m). To compare the average difference between the groups, a two-tailed Student’s *t*-test was used. *P* < 0.05 was regarded as a statistically significant difference.

## 3. Results

### 3.1. Identification of the SESN3 Proliferative Role in the Pre-Adipocyte

We constructed the pcDNA3.1-Sesn3-CDS vector to explore its proliferative role in pre-adipocytes, then transfected them into pre-adipocytes for 48 h and detected their proliferative efficiency by EdU incubation and CCK-8 detection. The confocal fluorescence detection result showed the positive EdU level of SESN3 transfection groups was down-regulated compared with control group (Figure 1A,B). Moreover, CCK-8 detection resulted in further identified SESN3-inhibited pre-adipocyte proliferation (Figure 1C).

Besides, we designed the specific small interference RNA of *Sesn3* (si-Sesn3), then transfected them into pre-adipocyte for 48 h. After EdU incubation, the confocal fluorescence detection result showed the positive EdU level of si-Sesn3 transfection group was upregulated compared with NC group (Figure 1D,E). The CCK-8 result showed that si-Sesn3 increased pre-adipocyte proliferation (Figure 1F).

Above results demonstrated that SESN3 inhibited pre-adipocyte proliferation. 

### 3.2. Sesn3 Inhibits Pre-Adipocyte Adipogenesis

Due to its suppression proliferative role in pre-adipocytes, we transfected the pcDNA3.1-Sesn3-CDS into pre-adipocytes to confirm its adipogenic role (Figure 2A). We found that except for the *PPARγ*, other adipogenic marker genes including *C/EBPα*, *C/EBPβ*, and *Fabp4* in Sesn3-overexpressed group were downregulated compared with the control group (Figure 2B). Pre-adipocytes were stimulated with DIM for 8 days after transfection, and we found that pre-adipocyte differentiation was suppressed in the pcDNA3.1-Sesn3-CDS-transfected group (Figure 2C,D). Moreover, si-Sesn3 was further transfected into the pre-adipocyte (Figure 2E), we found that *C/EBPα*, *C/EBPβ*, and *PPARγ* expression were upregulated in the si-Sesn3-transfected group (Figure 2F). After stimulating with DIM for 8 days, pre-adipocyte differentiation was improved in the si-Sesn3 transfection group (Figure 2G,H). Hence, these results identified that SESN3 inhibited pre-adipocyte adipogenesis.

### 3.3. SESN3 Inhibits SMAD3 to Reduce Its Suppressing Effect on ssc-miR-124 Transcription

According to the report, SESN3 has been demonstrated to inhibit SMAD3, thus suppressing the TGFβ-SMAD3 signal pathway [24]. Thus, we respectively transfected pcDNA3.1-Sesn3-CDS and si-Sesn3 into pre-adipocyte for 48 h, then detected the expression patterns of *Smad3*. We found that Sens3 inhibited *Smad3* expression, which was consistent with the report (Figure 3A–D). Our previous study identified that *mmu-miR-124* played an important role in 3T3-L1 adipogenesis [21]; hence, we predicted the *ssc-miR-124* promoter and further explored whether the SMAD3 regulated *ssc-miR-124* transcription by binding to promoter. We predicted 4 binding sites, and the dual-luciferase reporter assay system result showed that the predicted binding site 3 was the most suppressive efficiency one (Figure 3E,F). We further confirmed this result via the ChIP-PCR (Figure 3G). Besides, the pcDNA3.1-Smad3-CDS vector was transfected to pre-adipocyte, and the result showed that Smad3 suppressed the *ssc-miR-124* expression level (Figure 3H).

### 3.4. Identification of ssc-miR-124 Proliferative Role in the Pre-Adipocyte

To identify the role of *ssc-miR-124* in pre-adipocytes proliferation, we transfected them into pre-adipocytes for 48 h. The confocal fluorescence detection result showed that compared with NC group, the positive EdU level of *ssc-miR-124* mimicking the transfection group was downregulated (Figure 4A,B), and the CCK-8 detection result further proved that *ssc-miR-124* mimics inhibited pre-adipocyte proliferation (Figure 4C).

Besides, we also transfected the *ssc-miR-124* inhibitors into pre-adipocyte for 48 h. The confocal fluorescence detection result identified that the positive EdU level of *ssc-miR-124* inhibitors was upregulated compared with NC groups (Figure 4D,E). Moreover, the CCK-8 result further confirmed that *ssc-miR-124* inhibitors improved pre-adipocyte proliferation (Figure 4F).

The above results further demonstrated that *ssc-miR-124* impaired pre-adipocytes proliferation.

### 3.5. Ssc-miR-124 Inhibits Pre-Adipocyte Adipogenesis

To identify the adipogenic role of *ssc-miR-124* in porcine pre-adipocytes, we transfected *ssc-miR-124* mimics and s*sc-miR-124* inhibitors into porcine pre-adipocyte, respectively (Figure 5A,E), and detected the expression level of adipogenic marker genes, including *C/EBPα/β*, *PPARγ*, and *Fabp4*. The result showed that *ssc-miR-124* inhibited these genes’ expression (Figure 5B,F). To confirm the inhibitory role of *ssc-miR-124* in pre-adipocyte adipogenesis, we further stimulated pre-adipocyte with DIM for 8 days after transfecting *ssc-miR-124* mimics or inhibitors into pre-adipocyte for 48 h. The results identified that *ssc-miR-124* mimics inhibited pre-adipocyte differentiation (Figure 5C,D), and *ssc-miR-124* inhibitors promoted pre-adipocytes differentiation (Figure 5G,H).

To sum up, we found *ssc-miR-124* suppressed porcine pre-adipocytes adipogenesis.

### 3.6. Ssc-miR-124 Inhibits C/EBPα and GR by Targeting Their 3′UTR

*Ssc-miR-124* targets were predicted by 4 main target prediction tools. After the comprehensive analysis, we found 18 genes were shared among 4 prediction tools (Figure 6A), among them *C/EBPα* and *GR*. Hence, we further analyzed the *C/EBPα* and *GR* and respectively predicted 3 *ssc-miR-124* binding sites in *C/EBPα* gene 3′UTR and 2 *ssc-miR-124* binding sites in *GR* gene 3′UTR (Figure 6B). We constructed the pmirGLO vector with *C/EBPα* and *GR* gene 3′UTR which contained the predicted *ssc-miR-124* binding sites and their mutated binding sites (Figure 6B). The dual-luciferase reporter assay results identified that *ssc-miR-124* targeted *C/EBPα* and *GR* gene (Figure 6C,D). We further transfected *ssc-miR-124* mimics into pre-adipocyte for 48 h, and western bolting results showed that *ssc-miR-124* inhibited C/EBPα and the GR protein level (Figure 6E,F). Besides, we also transfected *ssc-miR-124* inhibitors into pre-adipocytes, and western bolting results identified that *ssc-miR-124* inhibitors increased C/EBPα and GR protein levels (Figure 6G,H).

The above results further proved that *ssc-miR-124* suppressed porcine *C/EBPα* and *GR* by targeting their 3′UTR, and thus inhibiting their protein expression.

## 4. Discussion

In general, Sestrin was regarded as a versatile anti-aging molecule, which was considered an important component of antioxidant defense, and was transcriptionally induced upon oxidative damage through diverse transcription factors, for example p53, Nrf2, AP-1, and FoxOs [23]. The excessive reactive oxygen species (ROS) was considered correlated with the aging so far, although an appropriate level of ROS was necessary for physiological homeostasis. Besides, other promoters of aging include nutritional overabundance and obesity. It is known that the mechanistic target of rapamycin complex 1 (mTORC1) is one of major regulators that mediates the nutritional effect on aging; meanwhile, including mammals, either genetic or pharmacological inhibition of mTORC1 extends longevity and health span in most organisms [30,31]. The Sestrin family contains three paralogues in vertebrates, they were Sestrin 1–3, respectively [32], specially, Sestrin 1 and Sestrin 2 have been demonstrated tp reduce ROS and suppress mTORC1 activity [33]. Therefore, the suppression of ROS or mTORC1 was treated as a therapy among fat accumulation, insulin resistance, muscle degeneration, cardiac dysfunction, mitochondrial pathologies and tumorigenesis that was led by genetic depletion of Sestrin in many animal models, for example, worms [34], flies [35], and mice [36]. However, Sestrin 1 and Sestrin 2 were highly conserved between their protein structure, while the structure of Sestrin 3 differed from other eukaryotic proteins, including Sestrin 1 and 2 [23]. By report, chronic activation of p70 S6 kinase (S6K) and mTORC1 in response to hypernutrition contributed to obesity-associated metabolic pathologies including hepatosteatosis and insulin resistance. Furthermore, the ablation of the Sestrin 2 and 3, especially Sestrin 2, provoked hepatic mTORC1-S6K activation and insulin resistance [36]. On the other hands, some reports have been identified that the Sesn2 induced by a high-fat diet in the liver and skeletal muscle of mice, whereas Sesn1 was decreased in liver, and Sesn3 was decreased in liver and adipose tissue. In the skeletal muscle of mice and the leg muscle biopsies of human diabetics, Sesn3 was increased [37,38], which would offer parts of explanation for the adipogenic phenotype we observed. However, the adipogenic role of Sestrin 3 among porcine pre-adipocyte differentiation and proliferation still needs to be further explored.

We first identified the macroscopically adipogenic role of SESN3, and we found SESN3 inhibited pre-adipocyte proliferation thus suppressed its adipogenesis (Figure 1 and Figure 2). Although cell proliferation and differentiation are usually incompatible within the same cell cycle, generally, it is necessary to stop proliferation then initiate differentiation. However, our results showed that Sesn3 inhibited pre-adipocyte proliferation and differentiation, which seemingly was not consistent with that. However, as the basic unit of adipose tissue, the lipid droplet enrichment of single adipocyte or increasing the number of adipocytes both lead to adipogenesis development. Meanwhile, we detected pre-adipocyte differentiation by transfecting vector or siRNA for 48 h, as well as after 8 days of induction, when we detected their proliferative effect was at the end of transfection. Hence, we consider that SESN3 regulated pre-adipocyte adipogenesis through inhibiting its proliferation effect, and then further suppressing differentiation. There are some reports that also concur this idea: For instance, the inhibition effect of HMGB2 and KDM5 for pre-adipocytes thus suppressed adipogenesis [39,40,41].

Besides, SESN3 has been demonstrated to be in involved TGFβ-Smads signaling by inhibiting the SMADs family at both protein and mRNA expression levels, thus protecting against diet-induced non-alcoholic steatohepatitis (NASH) in mice [24]. Thus, we further detected its inhibition for SMAD3 by respectively transfecting pcDNA3.1-Sesn3-CDS and si-Sesn3. Our results further identified that Sesn3 inhibited Smad3 within the porcine pre-adipocytes (Figure 3).

We demonstrated that *mmu-miR-124* played an important role in 3T3-L cell adipogenesis; however, the adipogenic molecular mechanism of *ssc-miR-124* for pigs still needs to be further identify [21]. We found 4 predicted binding sites of SMAD3 in the *ssc-miR-124* promoter region, and the results of the dual-luciferase reporter system and ChIP-PCR further identified that the predicted binding site3 was a positive site (Figure 3C–E). To identify the regulatory role of SMAD3 for *ssc-miR-124*, results showed that SMAD3 inhibited *ssc-miR-124* expression (Figure 3F).

To understand the adipogenic role of *ssc-miR-124* in pigs, we respectively detected its proliferative and differentiated effect for porcine pre-adipocyte (Figure 4 and Figure 5). As was our prediction, *ssc-miR-124* promoted pre-adipocyte proliferation, thus improving its differentiation, which was consistent with the SESN3 adipogenic role (Figure 1 and Figure 2), as well as being consistent with our previous report of *mmu-miR-124* regulation for 3T3-L1 cell [21].

Based on the post-transcriptional regulatory role of miRNA, we further explored ssc-miR-124 underlying molecular mechanisms. We identified two key adipogenic factors, GR and C/EBPα, which were targets of *ssc-miR-124*. As one of the most important adipogenic regulators, C/EBPα has been demonstrated interacting with PPARγ to regulate insulin signal pathway that playing the crucial role among adipogenesis [5,6,42,43]. For *GR*, also known as the *NR3C1* gene, it was a nuclear receptor that activated by ligand binding. Glucocorticoid (GC) was one of the key signals that were capable of inducing adipocyte differentiation, while glucocorticoid sensitivity of a cell largely depended on its GR levels [21,44,45].

Our results demonstrated that *ssc-miR-124* targeted *GR* 3′UTR and *C/EBPα* 3′UTR and inhibited their mRNA and protein levels, thus suppressing those two regulators from regulating adipogenesis among pre-adipocytes (Figure 6).

## 5. Conclusions

We identified that SESN3 inhibited SMAD3, thus improving *ssc-miR-124* expression, and finally suppressing two adipogenic vital factors, *GR* and *C/EBPα*, to regulate pre-adipocyte adipogenesis.

## Figures and Tables

**Figure 1 genes-12-01852-f001:**
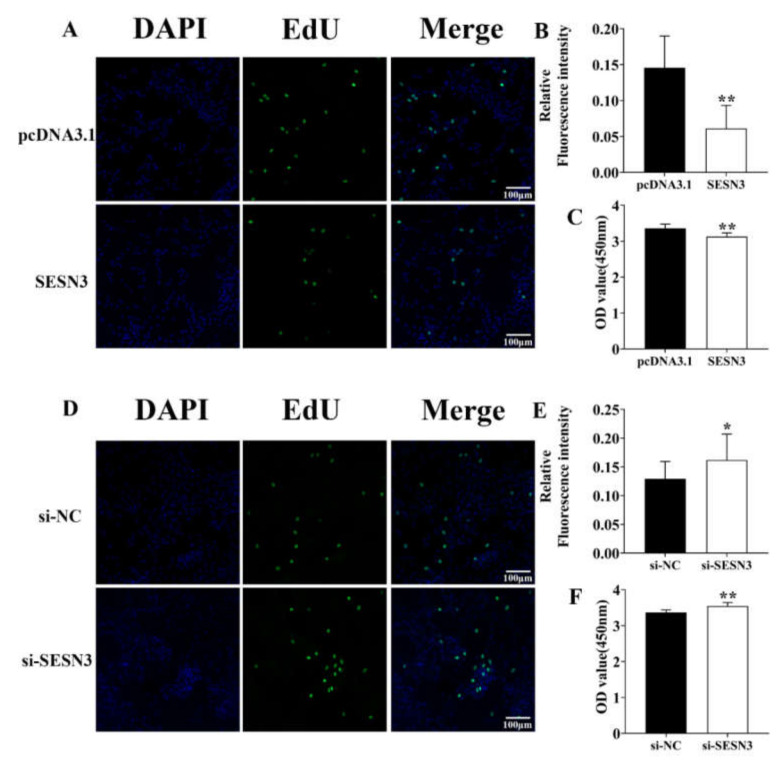
SESN3 inhibited pre-adipocyte proliferation. (**A**,**D**) Detection of pre-adipocyte positive EdU level after transfecting pcDNA3.1-Sesn3-CDS vector (**A**) or si-Sesn3 (**D**) for 48 h. The proliferating nuclei were stained green with EdU (50 μM) for 4 h, while the nuclei of all cells were stained blue with Hoechst33342 for 30 min. The scale bar was 100 μm. (**B**,**E**) Quantification of the relative fluorescence intensity of EdU-positive cells (EdU/DAPI) using ImageJ software. (**C**,**F**) The Cell Counting Kit-8 assay (CCK-8) detection after transfecting pcDNA3.1-Sesn3-CDS vector (**A**) or si-Sesn3 (**D**) for 48 h (n = 9 per group). The wavelength of CCK-8 detection was 450 nm. Data are presented as mean ± s.e.m. * means *P* < 0.05, ** means *P* < 0.01.

**Figure 2 genes-12-01852-f002:**
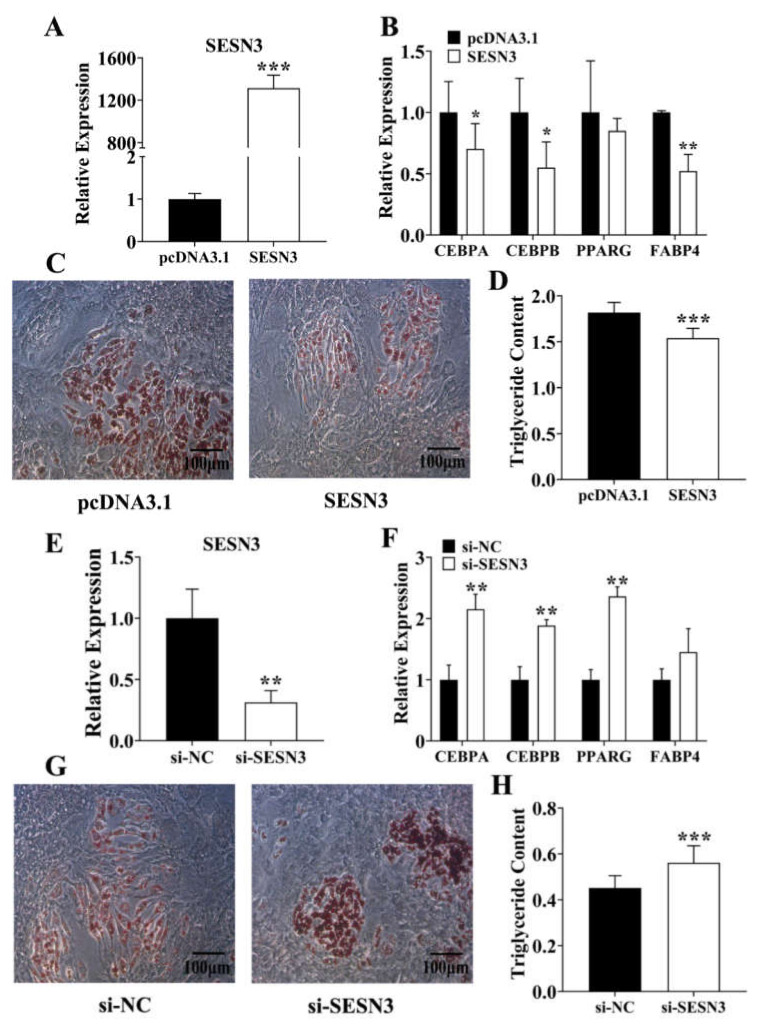
SESN3 inhibited pre-adipocyte adipogenesis. (**A**) *Sesn3* was overexpressed in pre-adipocytes after transfecting pcDNA3.1-Sesn3-CDS vector. (**B**) The expression of adipogenic marker genes after transfecting pcDNA3.1-Sesn3-CDS vector into pre-adipocyte for 48 h. (**C**,**D**) The pre-adipocyte oil red O staining (**C**) and quantitation of total triglyceride result (**D**) after transfecting pcDNA3.1-Sesn3-CDS vector. (**E**) *Sesn3* expression was inhibited after transfecting si-Sesn3. (**F**) The expression of adipogenic marker genes after transfecting si-Sesn3 into pre-adipocyte for 48 h. (**G**,**H**) The pre-adipocyte oil red O staining (**G**) and quantitation of total triglyceride result (**H**) after transfecting si-Sesn3. Data are presented as mean ± s.e.m. * means *P* < 0.05, ** means *P* < 0.01, and *** means *P* < 0.001.

**Figure 3 genes-12-01852-f003:**
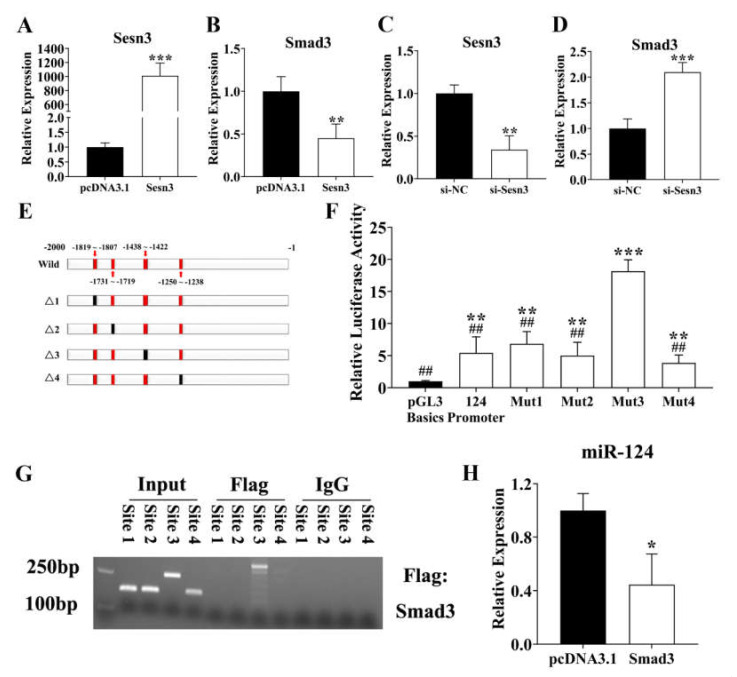
SMAD3 suppressed *ssc-miR-124* transcription by binding to its promoter. (**A**,**B**) Relative expression of *Sesn3* and *Smad3* after transfecting pcDNA3.1-Sesn3-CDS into pre-adipocyte. (**C**,**D**) Relative repression of *Sesn3* and *Smad3* after transfecting si-Sesn3 into pre-adipocyte. (**E**) The schematic diagram of SMAD3 binding sites in *ssc-miR-124* promoter. The red box meant wild binding sites, and black boxes were mutated ones. (**F**) Relative luciferase activity analysis in pre-adipocytes after pGL3 basic vector that contained the sequence with wild binding site or mutated one transfection. (**G**) ChIP-PCR detection of the binding of SMAD3 in *ssc-miR-124* promoter region. (**H**) The miRNA real-time PCR analysis of *ssc-miR-124* expression level in pre-adipocyte after transfecting pcDNA3.1-Smad3-CDS for 48 h. Data are presented as mean ± s.e.m. * means *P* < 0.05, ** means *P* < 0.01, and *** means *P* < 0.001, which shows the data compared with the control group, **##** represented significant difference between promoter Mut3 and other five groups, and means *P* < 0.01

**Figure 4 genes-12-01852-f004:**
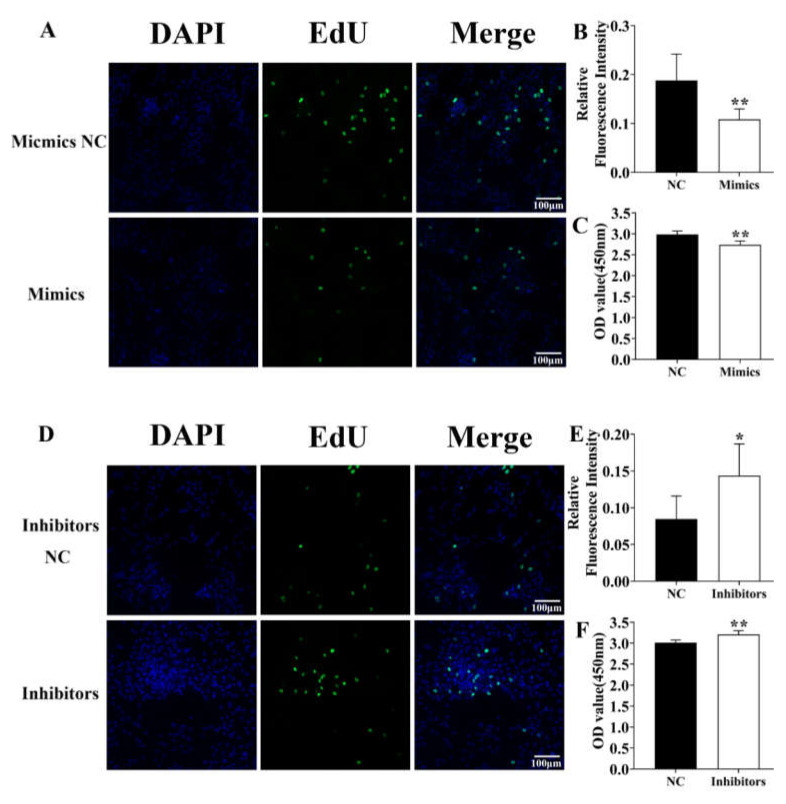
*Ssc-miR-124* inhibited pre-adipocyte proliferation. (**A**,**D**) Detection of pre-adipocyte positive EdU level after transfecting *ssc-miR-124* mimics (**A**) or *ssc-miR-124* inhibitors (**D**) for 48 h. The proliferating nuclei were stained green with EdU (50 μM) for 4 h, while the nuclei of all cells were stained blue with Hoechst33342 for 30 min. The scale bar was 100 μm. (**B**,**E**) Quantification of the relative fluorescence intensity of EdU-positive cells (EdU/DAPI) by ImageJ software. (**C**,**F**) The Cell Counting Kit-8 assay (CCK-8) detection after transfecting *ssc-miR-124* mimics (**A**) or *ssc-miR-124* inhibitors (**D**) for 48 h (n = 9 per group). The wavelength of CCK-8 detection was 450 nm. Data are presented as mean ± s.e.m. * means *P* < 0.05, ** means *P* < 0.01.

**Figure 5 genes-12-01852-f005:**
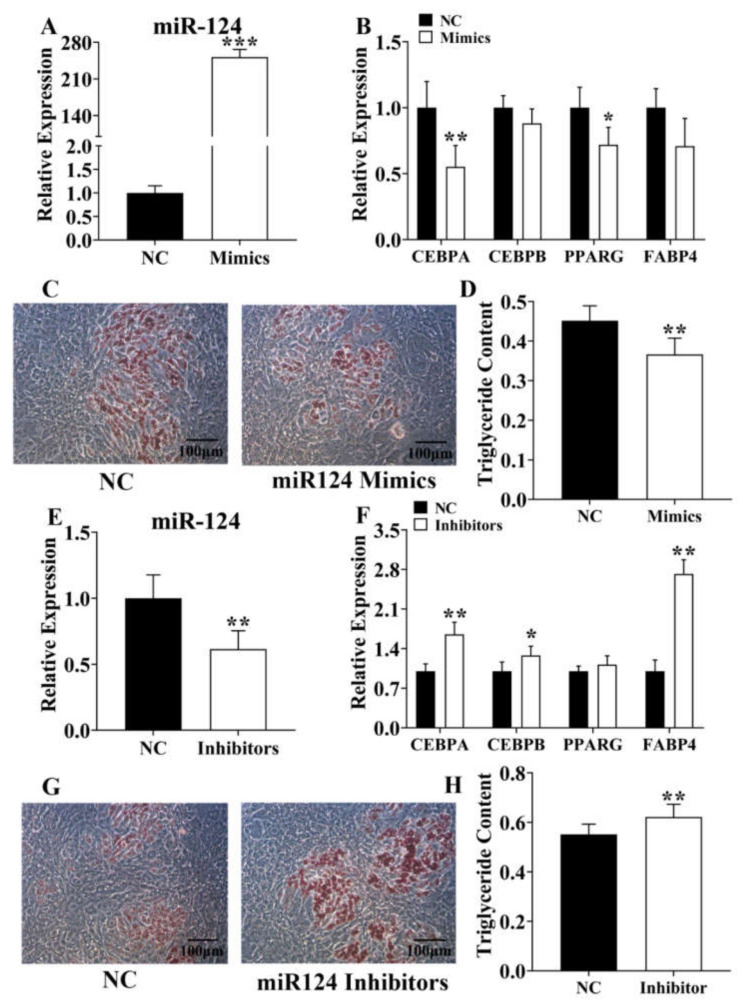
*Ssc-miR-124* inhibited pre-adipocyte adipogenesis. (**A**) *Ssc-miR-124* was overexpressed in pre-adipocyte by transfecting *ssc-miR-124* mimics. (**B**) The real-time analysis of adipogenic marker genes in pre-adipocyte after transfecting *ssc-miR-124* mimics for 48 h. (**C**,**D**) The pre-adipocytes oil red O staining (**C**) and quantitation of total triglyceride result (**D**) after transfecting *ssc-miR-124* mimics. (**E**) *Ssc-miR-124* was inhibited in pre-adipocyte after transfecting *ssc-miR-124* inhibitors. (**F**) The real-time analysis of adipogenic marker genes after transfecting *ssc-miR-124* inhibitors for 48 h. (**G**,**H**) The pre-adipocyte oil red O staining (**G**) and quantitation of total triglyceride result (**H**) after transfecting *ssc-miR-124* inhibitors. Data are presented as mean ± s.e.m. * means *P* < 0.05, ** means *P* < 0.01, and *** means *P* < 0.001.

**Figure 6 genes-12-01852-f006:**
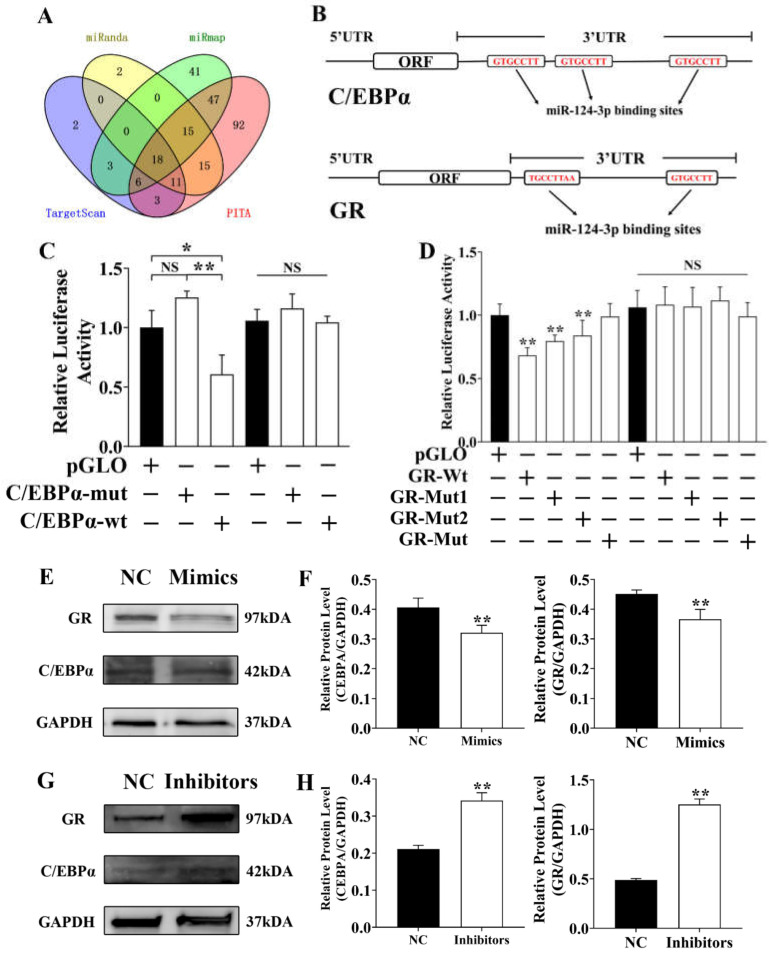
*Ssc-miR-124* suppressed *GR* and *C/EBPα* by targeting their 3′UTR. (**A**) The Venn diagram of *ssc-miR-124* targets by 4 prediction software. (**B**) The schematic diagram of *ssc-miR-124* miRNA response elements (MREs) of *C/EBPα* and *GR* genes. (**C**,**D**) Relative luciferase activity of constructed pmirGLO vector that contained the *C/EBPα* 3′UTR and *GR* 3′UTR, which respectively contained *ssc-miR-124* wild or mutated binding site sequences. (**E**,**F**) Western blotting result of C/EBPα and GR in pre-adipocyte after transfecting *ssc-miR-124* mimics for 48 h (**E**), and the relative quantification of their WB results used ImageJ software (**F**). (**G**,**H**) Western blotting results of C/EBPα and GR in pre-adipocyte after transfecting ssc-miR-124 inhibitors for 48 h (**G**), and the relative quantification of their WB results using ImageJ software (**H**). Data are presented as mean ± s.e.m. * means *P* < 0.05, ** means *P* < 0.01.

## Data Availability

The data that support the findings of this study are available from the corresponding author upon reasonable request.

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
