# Peer review of "SESN3 Inhibited SMAD3 to Relieve Its Suppression for MiR-124, Thus Regulating Pre-Adipocyte Adipogenesis"

_genes, 2021, doi:10.3390/genes12121852_

Round 1

Reviewer 1 Report

General comment- This is a clearly presented and well written paper from Dr Chen's group. In this manuscript, authors examined the regulatory role of sestrin-3 in adipogenesis. Sestrin 3 inhibited SMAD3 to relieve its suppression for miRNA-124 which inhibits pre-adipocyte proliferation and adipogenesis via suppressing C/EBPaand GR. Using overexpressed SESN3 and siRNA against SESN3 transfection in porcine adipocytes revealed role of SESN3 in adipogenesis. The proposed study is interesting but I have the following comments and concern.

Major Comments-

  1. Authors did not discuss the role of sestrin 3 in mouse or human primary adipocytes or cell lines. What is the status of sestrin 3 regulation in these cell types? .
  2. Does sestrin 3 expression changes in human conditions such as obesity/NAFLD or diabetes etc? What is known about sestrin 3 in literature in these settings. If possible, explain.

Minor Comments-

  1. Spelling mistake in abstract- Adiogenesis or adipogenesis (line 16) ?

  1. Fig 3. and Fig 5. mentions symbol *** in subpanel but this symbol did not mention or written in fig. legends.

Author Response

General comment- This is a clearly presented and well written paper from Dr. Chen's group. In this manuscript, authors examined the regulatory role of sestrin-3 in adipogenesis. Sestrin 3 inhibited SMAD3 to relieve its suppression for miRNA-124 which inhibits pre-adipocyte proliferation and adipogenesis via suppressing C/EBPa and GR. Using overexpressed SESN3 and siRNA against SESN3 transfection in porcine adipocytes revealed role of SESN3 in adipogenesis. The proposed study is interesting but I have the following comments and concern:

Major Comments-

  1. Authors did not discuss the role of sestrin 3 in mouse or human primary adipocytes or cell lines. What is the status of sestrin 3 regulation in these cell types?

Author Reply: Thank you. Currently, for the role of Sesn3 in obesity or NAFLD or other lipid mechanism usually was considered to involve mTROC signaling, to be specific, Sestrin family strongly inhibited mTORC1 signaling, while promoted mTORC2-dependent AKT phosphorylation in mice cell lines or Drosophila tissues. Meanwhile, the chronic activation of mTORC1 and S6K signaling lead to insulin resistance, which was an important adipogenic regulatory mechnism (Ho Allison et al, Trends in biochemical sciences, 2016). That was the most popular explain for Sesn3 adipogenic role.

  1. Does sestrin 3 expression changes in human conditions such as obesity/NAFLD or diabetes etc? What is known about sestrin 3 in literature in these settings. If possible, explain.

Author Reply: Thank you. We have supplemented the explanation into the Discussion. Some reports have been identified that the Sesn2 in duced by high-diet in the liver and skeletal muscle of mice, whereas Sesn1 was decreased in liver, Sesn3 was decreased in liver and adipose tissue. In the skeletal muscle of mice and the leg muscle biopsies of human diabetics, Sesn3 was both increased. (Dong et al, Expert opinion on therapeutic targets, 2015; Kim et al, Annual review of physiology, 2021). For their explanation still focused on the mTROC signaling regulation.

Minor Comments-

  1. Spelling mistake in abstract- Adiogenesis or adipogenesis (line 16)?

Author Reply: Thank you, we have revised the misspelling adiogenesis to adipogenesis, and fully checked the manuscript to avoid the same mistake.

  1. Fig. 3 and Fig. 5 mentions symbol *** in subpanel but this symbol did not mention or written in fig. legends.

Author Reply: Thank you. We have supplemented the explanation of symbol *** in Fig.3 and Fig.5 legends.

Reviewer 2 Report

I believe you have not finished your sentence in line 92.

Did you finish your sentence in lines 183-184?

Subsections 3.2, 3.3, 3.4, 3.5 and 3.6 seem to be a little too descriptive and analytical. I would suggest shortening the paragraphs in the Results section, they should only state your results without any interpretation and scientific background. Some parts of these paragraphs should be placed in the Materials and Methods section, while sentences with a broader description with possible implications of your findings should be placed in the Discussion section. In addition, I would suggest moving the title of the subsection 3.3 to the next page.

In line 373 I would suggest you elaborate on which reports agreed with your findings and what they were about.

There is an adequate number of figures to allow for a better understanding of your research, though Figure 6 is a little difficult to grasp because of all the different graphs, signs and descriptions so close together.

The methods used in this study are correctly and thoroughly described, allowing for replication of this study.

The manuscript needs a major language correction, preferably by a native English speaker.

Author Response

I believe you have not finished your sentence in line 92.

Author Reply: Thank you, we have supplemented the information in line 92.

Did you finish your sentence in lines 183-184?

Author Reply: Thank you, we have revised the sentence.

Subsections 3.2, 3.3, 3.4, 3.5 and 3.6 seem to be a little too descriptive and analytical. I would suggest shortening the paragraphs in the Results section, they should only state your results without any interpretation and scientific background.

Author Reply: Thank you. We have revised and bowdlerized the redundant description and analysis in the results subsections.

Some parts of these paragraphs should be placed in the Materials and Methods section, while sentences with a broader description with possible implications of your findings should be placed in the Discussion section. In addition, I would suggest moving the title of the subsection 3.3 to the next page.

Author Reply: Thank you. We have fully checked the manuscript and revised the sentences and subsections followed the advices.

In line 373 I would suggest you elaborate on which reports agreed with your findings and what they were about.

Author Reply: Thank you. We have revised the sentence followed the advice.

There is an adequate number of figures to allow for a better understanding of your research, though Figure 6 is a little difficult to grasp because of all the different graphs, signs and descriptions so close together.

The methods used in this study are correctly and thoroughly described, allowing for replication of this study.

The manuscript needs a major language correction, preferably by a native English speaker.

Author Reply: Thank you. We have fully checked and revised the manuscript.